# SGLT2 Inhibitors in Type 2 Diabetes Mellitus and Heart Failure—A Concise Review

**DOI:** 10.3390/jcm11061470

**Published:** 2022-03-08

**Authors:** Daria M. Keller, Natasha Ahmed, Hamza Tariq, Malsha Walgamage, Thilini Walgamage, Azad Mohammed, Jadzia Tin-Tsen Chou, Marta Kałużna-Oleksy, Maciej Lesiak, Ewa Straburzyńska-Migaj

**Affiliations:** 1Ist Department of Cardiology, Poznan University of Medical Sciences, 61-848 Poznan, Poland; jadzia.chou@gmail.com (J.T.-T.C.); marta.kaluzna@wp.pl (M.K.-O.); maciej.lesiak@skpp.edu.pl (M.L.); ewa.straburzynska-migaj@skpp.edu.pl (E.S.-M.); 2Cardiology Department, Lancashire Teaching Hospitals NHS Foundation Trust, Preston PR2 9HT, UK; natashaahmed@doctors.org.uk; 3Cardiology Department, University Hospitals Coventry and Warwickshire NHS Trust, Coventry CV2 2DX, UK; hamza_tariq@hotmail.co.uk; 4Center for Medical Education in English, Poznan University of Medical Sciences, 60-512 Poznan, Poland; malsha.walgamage@gmail.com (M.W.); thilini.walgamage@gmail.com (T.W.); azadmohammed158@gmail.com (A.M.)

**Keywords:** sodium-glucose co-transporter 2 inhibitors, sodium-glucose co-transporter 2, diabetes mellitus type 2, heart failure

## Abstract

The incidence of both diabetes mellitus type 2 and heart failure is rapidly growing, and the diseases often coexist. Sodium-glucose co-transporter 2 inhibitors (SGLT2i) are a new antidiabetic drug class that mediates epithelial glucose transport at the renal proximal tubules, inhibiting glucose absorption—resulting in glycosuria—and therefore improving glycemic control. Recent trials have proven that SGLT2i also improve cardiovascular and renal outcomes, including reduced cardiovascular mortality and fewer hospitalizations for heart failure. Reduced preload and afterload, improved vascular function, and changes in tissue sodium and calcium handling may also play a role. The expected paradigm shift in treatment strategies was reflected in the most recent 2021 guidelines published by the European Society of Cardiology, recommending dapagliflozin and empagliflozin as first-line treatment for heart failure patients with reduced ejection fraction. Moreover, the recent results of the EMPEROR-Preserved trial regarding empagliflozin give us hope that there is finally an effective treatment for patients with heart failure with preserved ejection fraction. This review aims to assess the efficacy and safety of these new anti-glycemic oral agents in the management of diabetic and heart failure patients.

## 1. Introduction

Diabetes mellitus type 2 (T2DM) is a rapidly growing metabolic disorder affecting over 400 million people worldwide [1]. The increasing prevalence is correlated with the direct global increase in obesity, which is a significant risk factor of T2DM. The pathophysiology of T2DM is primarily related to insulin resistance, which leads to hyperglycemia and a gradual decrease in the β-cells’ ability to produce insulin. Furthermore, pancreatic α-cell dysfunction, increased hepatic glucose output, impaired incretin effect, increased renal glucose reabsorption, and neurotransmitter dysregulation in the central nervous system also contribute to T2DM development [2]. The disease is strongly associated with both micro- and macrovascular complications. Therefore, cardiovascular (CV) diseases, especially heart failure (HF), are the greatest burden on healthcare expenditure and have the highest impact on mortality within the diabetic population [3]. The Candesartan in Heart failure—Assessment of mortality and Morbidity (CHARM) study demonstrated that a 1% increase in glycosylated hemoglobin A1c (HbA1c) is associated with a 25% increase in the risk for CV events or death in T2DM patients [4]. The close association between diabetes mellitus and HF is a result of the detrimental effect of pivotal pathogenic factors: chronic glucotoxicity and lipotoxicity, as well as altered insulin signaling. Myocardial structural and functional derangement is a consequence of oxidative stress, increased formation of advanced glycation end products, altered intracellular calcium handling, endothelial dysfunction, and inflammation [5].

According to the most recent 2021 guidelines published by the European Society of Cardiology (ESC), HF is a clinical syndrome comprising essential symptoms—for example, dyspnea, fatigue, and ankle swelling—that may be associated with signs—elevated jugular venous pressure, pulmonary crackles, or peripheral edema—due to structural and/or functional abnormality of the heart resulting in elevated intracardiac pressures and/or insufficient cardiac output at rest and/or during physical activity. Most commonly, HF is a result of myocardial systolic and/or diastolic dysfunction; however, pathology of the valves, endocardium, pericardium, or arrhythmias may also contribute to the disease [6]. HF is common in T2DM patients, and its current prevalence is estimated at >64 million cases [7]. This highlights not only the importance of tight glycemic control for T2DM patients but also CV management.

While metformin (MET) seems to exhibit cardioprotective effects, other traditional antidiabetic drugs have neutral or even harmful impacts on CV outcomes [8]. It is well established that saxagliptin and alogliptin should be avoided in the HF population, whereas pioglitazone is unequivocally contraindicated [9]. Although glucagon-like peptide-1 (GLP-1) receptor agonists reduce the risk of myocardial infarction (MI), stroke, and CV death in patients with T2DM, they are not recommended for the prevention of HF events [6].

Sodium-glucose co-transporter 2 inhibitors (SGLT2i) are a new antidiabetic drug class that mediates epithelial glucose transport at the renal proximal tubules, inhibiting glucose absorption—resulting in glycosuria—and therefore improving glycemic control [10]. SGLT2i have also demonstrated CV benefits, especially in the treatment of HF. Canagliflozin (CANA) exhibited efficient glycemic control while reducing HbA1c, becoming the first Food and Drug Administration (FDA)-approved SGLT2i in 2013. Moreover, the Canagliflozin Cardiovascular Assessment Study (CANVAS) demonstrated the potential of CANA to reduce the risk of CV complications in T2DM patients, including non-fatal stroke, non-fatal MI, and HF management, further highlighting the cardioprotective qualities of SGLT2i and their role in T2DM treatment, in conjunction with first-line treatment with MET [11].

The FDA and the European Medicines Agency (EMA) have currently approved four oral SGLT2i agents for T2DM treatment: CANA, dapagliflozin (DAPA), ertugliflozin (ERTU), and empagliflozin (EMPA). Ipragliflozin, luseogliflozin, and tofogliflozin have been approved in Japan. Remogliflozin etabonate was first commercially launched in India. Research on sergliflozin etabonate was discontinued after phase II trials. Notably, sotagliflozin (SOTA) is a dual SGLT1/SGLT2 inhibitor. The FDA refused to approve its use in combination with insulin for diabetes mellitus type 1 (T1DM), and phase III trials on SOTA in patients with T2DM and HF were regrettably and untimely terminated due to financial reasons and the COVID-19 pandemic.

The 2021 ESC guidelines recommended two SGLT2i, DAPA and EMPA, as first-line treatment for HF patients with reduced left ventricular ejection fraction (LVEF), namely HF with reduced ejection fraction (HFrEF), along with other recommended first-line agents regardless of the presence of diabetes, unless contraindicated or not tolerated (class I). Furthermore, CANA, DAPA, EMPA, ERTU, and SOTA are recommended in T2DM patients at risk of CV events in order to reduce hospitalizations due to HF, major CV events, end-stage renal disease (ESRD), as well as CV death (also class I) [6].

This review aims to assess the efficacy and safety of these new anti-glycemic oral agents in the management of diabetic and HF patients. 

## 2. Sodium-Glucose Co-Transporters

Under normal physiological conditions, the kidney has the potential to reabsorb glucose at a rate of 375 mg/min, while the glomeruli filter glucose at a rate of 125 mg/min [10,12]. Thus, the kidney can theoretically filter 300 mg/dL of glucose. However, blood glucose levels exceeding 180 mg/dL result in glycosuria. The difference between the hypothetical threshold of 300 mg/dL and the physiological threshold of 180 mg/dL is defined as splay [10,12]. Splay is determined by any morphological or functional differences in the glomeruli [13]. 

The renal proximal tubules are essential in the filtration and reabsorption of glucose, and they are responsible for the absence of glucose in urine [12,14]. The proximal tubules are composed of three segments: S1, S2, and S3, and they express co-transporters essential in facilitating glucose transport into the renal tubule [12,15,16]. Sodium-glucose co-transporters (SGLTs) are channel proteins that retain glucose against the concentration gradient [17,18,19]. There are two types of SGLTs: SGLT2, which are located proximally in the S1 segment, and SGLT1, which can be found distally in the S2 and S3 segments [12]. Thus, the S1 segment of the proximal tubules will filter a higher glucose concentration, which is then reabsorbed by SGLT2. Alternatively, the distal location of SGLT1 suggests that they have high affinity but are low-capacity glucose transporters (Figure 1) [9,20]. SGLT1 have a broader distribution, including the kidneys, intestines, heart, lungs, and skeletal muscles [20]. On the other hand, SGLT2 are limited to the kidney, pancreatic α-cells, and the cerebellum. SGLT2 are responsible for 80–90% of glucose reabsorption, while SGLT1 account for the remaining 10–20% [18,21]. 

## 3. Mechanism of Action of Sodium-Glucose Co-Transporters Inhibitors

A T2DM patient is in glycemic deprivation, which causes the kidneys to retain more glucose. As a result, glucose reabsorption capacity and the glucosuria threshold are increased [12,17]. Furthermore, the lack of glucose delivery to cells leads to the upregulation of SGLT2 for increased retention [13]. 

SGLT2i are oral antidiabetic drugs that competitively bind SGLT2 proteins in the S1 segments of proximal renal tubules. The subsequent inhibition of co-transporters leads to decreased glucose reabsorption and increases excretion of glucose into the urine [18]. Along with inducing glycosuria, they also stimulate osmotic diuresis and natriuresis via direct and indirect mechanisms [13,17]. As a result, they achieve the desired effect of decreasing plasma glucose levels. In contrast to other antidiabetic medications, SGLT2i are insulin-independent. Pancreatic β-cells are not overstimulated to secrete excessive insulin; thus, there is a lower risk of hypoglycemia. The SGLT2i mechanism is based solely on the renal glomerular tubular function. Therefore, reduced renal function will lead to decreased effectiveness of SGLT2i [22].

SGLT2i promote osmotic diuresis and natriuresis and therefore may reduce the preload [23,24,25]. Furthermore, gliflozin’s pleiotropic vascular effects result in vasodilation and reduction in the afterload [26,27,28,29]. Improvement of endovascular function due to indirect attenuation of oxidative stress and direct inhibition of pro-inflammatory mediators is also a major benefit [30]. Other studies have found that SGLT2i enhance myocardial metabolism, inhibit the sodium-hydrogen exchanger 1 isoform in the myocardium, and reduce calmodulin-dependent kinase II activity, which increases mitochondrial calcium levels and improves cardiac efficiency [31,32,33]. Potential beneficial outcomes of SGLT2i also include improvement of renal function, reduction in adipose tissue-mediated inflammation and sympathetic overdrive, modulation of the intrarenal renin–angiotensin system, as well as increase in erythropoietin levels and upregulation of provascular progenitor cells [34,35,36,37,38,39]. The protective effects of gliflozin on other organs are especially important, as the natural history of HF leads to their progressive involvement and dysfunction. Myocardial and systemic effects of SGLT2i are presented in Figure 2.

## 4. Clinical Pharmacology

### 4.1. Affinity

The affinity of different SGLT2i to SGLT2 varies. EMPA has the highest affinity to SGLT2 over SGLT1 (2500-fold), with the second being ERTU (2235-fold), followed by DAPA (1200-fold) and lastly CANA (200-fold). The difference in affinity may explain the differences in therapeutic and adverse effects of SGLTi [40].

### 4.2. Dosages

The starting and maintenance dosing of SGLT2i and their dosing schedules are summarized in Table 1 [41]. SGLT2i dosing, depending on the glomerular filtration rate, is presented in Table 2 [42].

## 5. Canagliflozin

CANA was approved in 2013 by the FDA following the results of the CANagliflozin Treatment and Trial Analysis (CANTATA) studies. Significant reduction in HBA1c and weight was demonstrated regardless of CANA use in monotherapy or as a second- or third-line agent [43,44].

### 5.1. Canagliflozin in Monotherapy

Stenlöf et al. [44] conducted a 26-week randomized controlled trial (RCT) in which 584 subjects received once daily 100 or 300 mg of CANA or placebo. HbA1c of patients treated with CANA 100 mg and 300 mg significantly decreased by 0.77% and 1.03%, respectively by the end of the trial. In contrast, HbA1c in the placebo group increased by 0.14%. Secondary endpoints such as fasting blood glucose (FPG), 2 h postprandial glucose, systolic blood pressure (SBP), and percent change in body weight with both doses of CANA were significantly reduced compared with placebo. Additionally, high-density lipoprotein cholesterol (HDL-C) levels increased from baseline in the CANA groups compared with placebo (*p* < 0.01). The study was extended to 52 weeks, and CANA’s impact on glycemic control and weight reduction was sustained [45].

### 5.2. Canagliflozin in Combination with Metformin

Wilding et al. [46] conducted a 26-week RCT, which was subsequently extended for another 26 weeks in 469 subjects with T2DM who were inadequately treated with MET and sulfonylurea (SU). Patients additionally received CANA at doses of 100 or 300 mg, or placebo. The primary endpoint was change in HbA1c at week 26. Secondary endpoints included change in HbA1c at week 52, and the following were assessed at weeks 26 and 52: proportion of patients who achieved HbA1c <7.0%, change in FPG and SBP, as well as percent change in body weight, HDL-C, and triglycerides. At week 26, HbA1c was significantly reduced in patients receiving CANA 100 and 300 mg compared to placebo (−0.85%, −1.06%, and −0.13%, respectively; *p* < 0.001). At week 52, HbA1c reduction was sustained (−0.74%, −0.96%, and 0.01%, respectively). Both doses of CANA also reduced weight and FPG levels in subjects at week 26 and week 52 [46].

Rosenstock et al. similarly demonstrated CANA’s superiority in reducing HbA1c when given in combination with MET. All treatment-naïve T2DM patients achieved HbA1c <7% when given both CANA and MET [47].

Yang et al. [48] performed a meta-analysis of six RCTs, assessing the efficacy and tolerability of CANA at doses 100 and 300 mg added to treatment with MET in T2DM patients. Compared to placebo, patients receiving CANA had an absolute reduction in HbA1c (−0.66% (−0.72%, −0.61%)). It was concluded that patients in both CANA groups were more likely to achieve the target HbA1c. FPG levels of the treatment groups were significantly reduced compared with placebo (−1.49 mmol/L for 100 mg/day and −1.80 mmol/L for 300 mg/day). CANA also demonstrated significant superiority in body weight improvement compared with placebo (−2.09% 100 mg/day and −2.66% for 300 mg/day) [48].

### 5.3. Canagliflozin vs. Sitagliptin as an Add-On Therapy to Metformin

T2DM patients inadequately controlled with MET additionally received 100 or 300 mg of CANA, 100 mg sitagliptin, or placebo. At week 52, there was no significant difference between either dose of CANA in lowering HbA1c. Moreover, sitagliptin was shown to be inferior at lowering HbA1c in comparison to 300 mg CANA (−0.88%, −0.73%, respectively). Both doses of CANA significantly reduced body weight, FPG, and SBP compared to sitagliptin (*p* < 0.001 for all) [49]. 

### 5.4. Canagliflozin in Combination with Metformin and Pioglitazone

In an RCT conducted by Forst et al. [50], T2DM patients ineffectively controlled with MET and pioglitazone additionally received 100 or 300 mg of CANA or placebo for 26 weeks. After this period, the placebo group switched to 100 mg of sitagliptin for another 26 weeks. At week 26, CANA 100 and 300 mg significantly lowered HbA1c (−0.89%, −1.03%) in comparison to placebo (0.26%; *p* < 0.001). This trend was continued at week 52 (−0.92% and −1.03% for 100 and 300 mg of CANA, respectively). Compared to placebo, patients receiving 100 and 300 mg of CANA lost a mean weight of −2.5 kg and −3.5 kg, respectively. Furthermore, FPG and SBP levels were significantly lowered at week 26 (*p* < 0.05 for both), with reductions maintained at week 52. Although the rates of adverse events (osmotic diuresis-related and genital mycotic infections) were low, they were significantly higher in CANA groups [50].

### 5.5. Canagliflozin in Combination with Insulin

Inagaki et al. investigated patients with T2DM inadequately controlled by insulin, diet, and physical activity, administering either a placebo or 100 mg CANA in a double-blind study in Japan. At week 16, the patients taking CANA had significantly reduced HbA1c baseline levels (−0.97) compared to the placebo group (0.13). Similar correlations were observed in relation to FPG (−34.1 vs. −1.4, CANA and placebo, respectively) and body weight (−2.13 vs. 0.24, CANA and placebo, respectively). On the other hand, HDL-C increased (3.3 ± 1.0 vs. −0.5 ± 1.0 mg/dL) [51].

## 6. Dapagliflozin

Phase III clinical trials on DAPA took place in 2010 with the FDA approving the drug in 2014.

### 6.1. Dapagliflozin in Monotherapy 

A study by Ferrannini et al. [52] compared three doses of DAPA: 2.5, 5, and 10 mg with placebo. Patients with T2DM inadequately controlled by diet and exercise were divided into two groups. The first group (HbA1c 7–10%) received once-daily placebo or one of the three doses of DAPA. These patients were categorized further into main or exploratory cohorts, with the main cohort receiving the dose in the morning and the exploratory cohort receiving it in the evening. The second group (HbA1c 10.1–12%) did not receive a placebo and was given morning doses of DAPA of either 5 or 10 mg daily. The mean change of HbA1c at week 24 in the placebo group was −0.23%, whereas DAPA at doses 2.5, 5, and 10 mg caused a mean HbA1c change of −0.58%, −0.77%, and −0.89%, respectively. The dosing time was not statistically significant. The study demonstrated the potential of DAPA to lower hyperglycemia in treatment-naïve patients with newly diagnosed T2DM [52].

### 6.2. Dapagliflozin and Metformin

Bailey et al. [53] conducted a 24-week study in 80 different sites around North and South America. Despite receiving the maximum dose (>1500 mg) of MET, these subjects had inadequately controlled glycemia. Doses of 2.5 mg, 5 mg, and 10 mg of DAPA were compared with a placebo. The baseline HbA1c saw greater reductions in the DAPA groups: 2.5 (*p* = 0.0002), 5, and 10 (both *p* < 0.0001) compared to placebo. Furthermore, the primary endpoint was met: more than 40% of patients taking 10 mg of DAPA achieved an HbA1c <7.0 after the 24 weeks. FPG was statistically decreased in DAPA doses of 5 and 10 mg [53].

The double-blind, three-arm, 24-week study by Henry et al. [54] compared the effects of DAPA and MET as monotherapies as well as in combination together for treatment-naïve patients. The study was conducted as two trials depending on DAPA doses (5 or 10 mg). MET doses were up-titrated to 2 g. Enrolled patients had baseline HbA1c 7.5–12%. Combination therapy led to the greater HbA1c reductions in both trials. Patients receiving 5 mg of DAPA and MET had an HbA1c reduction of −2.05. Those who were administered only 5 mg of DAPA or MET had a reduction of −1.19 and −1.35, respectively (*p* < 0.0001). Analogously, patients receiving 10 mg of DAPA and MET or only 10 mg DAPA or MET had HbA1c reduction of −1.98, −1.45, and −1.44, respectively (*p* < 0.0001). Combination therapy was also superior to monotherapy in the reduction in FPG (*p* < 0.0001) and superior to therapy with MET in the reduction in weight (*p* < 0.0001). Therapy with DAPA and MET was generally well-tolerated and effective in reducing HbA1c, FPG, and weight [54].

### 6.3. Dapagliflozin vs. Glipizide as an Add-On Therapy to Metformin

A 52-week study by Nauck et al. [55] on diabetic patients with inadequate glycemic control despite MET monotherapy compared an add-on of either DAPA (n = 406) or glipizide (n = 408). The drugs were up-titrated over 18 weeks based on tolerability and glycemic response, using up to 10 mg of DAPA or 20 mg of glipizide. Both DAPA and glipizide reduced mean HbA1c by −0.52%. Despite similar glycemic efficacy, DAPA was superior to glipizide in terms of weight reduction and lower rate of hypoglycemia [55].

## 7. Empagliflozin

The FDA approved EMPA for T2DM treatment in August 2014 [23].

### 7.1. Empagliflozin in Monotherapy

The EMPA-REG MONO trial (2010–2012) assessed the long-term efficacy and safety of EMPA monotherapy in 899 T2DM treatment-naïve patients. Patients were given EMPA at doses of 10 or 25 mg, with placebo and 100 mg of sitagliptin (once daily for 24 weeks in a 1:1:1:1 ratio) used as an active comparator. The adjusted mean differences in change from baseline HbA1c compared with placebo were −0.74% and −0.85% for 10 mg and 25 mg of EMPA, respectively. Sitagliptin showed a change of −0.73%. Patients with higher glycemia achieved a greater reduction in HbA1c when treated with EMPA. The change from baseline in FPG was −1.08 and −1.36 for EMPA at doses 10 and 25 mg, respectively, whereas sitagliptin did not decrease FPG as significantly (−0.38). EMPA was generally well tolerated [56].

### 7.2. Empagliflozin as an Add-On Therapy to Metformin

Häring et al. [57] investigated the effects of EMPA as an add-on to MET treatment in a 24-week study on patients inadequately controlled with 1500 mg of MET. A total of 637 patients received once daily 10 or 25 mg of EMPA or placebo in addition to current MET therapy. At week 24, the adjusted mean changes from baseline HbA1c were −0.57% and −0.64% for EMPA at doses of 10 and 25 mg, respectively. Patients also had a reduction in FPG: 0.35, −1.11, and −1.24 for placebo, 10 mg EMPA, and 25 mg EMPA, respectively [57]. This study concluded that EMPA was a safe and viable add-on to MET.

### 7.3. Empagliflozin as an Adjunctive to Therapy with Insulin in Diabetes Mellitus Type 1

Rosenstock et al. [58] conducted the Empagliflozin as Adjunctive to inSulin therapy (EASE) program, consisting of EASE-2 and EASE-3 trials, to evaluate the safety and efficacy of EMPA in 1707 patients with T1DM. The EASE-2 trial compared EMPA at doses of 10 or 25 mg, with placebo, whereas EASE-3 investigated a uniquely low dose of EMPA of 2.5 mg. At week 26, the largest observed mean placebo-subtracted HbA1c reductions were −0.25%, −0.54%, and −0.53% for 2.5, 10, and 25 mg of EMPA, respectively (*p* < 0.0001). EMPA improved glycemic control in T1DM without increasing hypoglycemic events; however, the rates of ketoacidosis were increased with 10 and 25 mg of EMPA [58].

## 8. Adverse Effects 

Adverse effects such as genitourinary tract infections, diabetic ketoacidosis, hypoglycemia, hypotension, bone fractures, and lower limb amputation are associated with SGLT2i (Figure 3). 

### 8.1. Genitourinary Tract Infections

Diabetic patients with poorly controlled glycemia have a higher risk of genital infections. This is due to the high concentration of glucose in the genitourinary tract as well as impaired immunity in the form of deranged neutrophil function, depressed antioxidant system, and a dysfunctional humoral response [59,60]. The incidence rate of urinary tract infections among T2DM patients in the UK General Practice Research Database was 46.9/1000 person-years compared with only 29.9/1000 person-years for patients without T2DM [61].

The SGLT2i mechanism of action further increases glucose concentration in the genitourinary tract, which increases the risk of infections [62]. This leads to a decrease in quality of life, later discontinuation of therapy and, consequently, poor glycemic control.

A large real-world retrospective cohort study by Dave et al. showed an approximately three-fold increased risk of genital infections during SGLT2i therapy compared to dipeptidyl peptidase-4 (DPP-4) inhibitors and GLP-1 receptor agonists [63]. Furthermore, a meta-analysis by Liu et al. showed that all SGLT2i have a significantly higher risk of urinary tract and genital infections; however, in case of DAPA, the relationship is dose-dependent [64]. For this reason, diabetic patients taking SGLT2i must be educated in maintaining a high standard of personal hygiene and adequate hydration to prevent infections. 

### 8.2. Diabetic Ketoacidosis

A documented side effect of SGLT2i therapy is euglycemic diabetic ketoacidosis, and as a result, the FDA discourages their use in T1DM patients. Glucose loss via SGLT2i-induced urination leads to decrease in insulin levels. Glucagon levels subsequently increase, leading to SGLT2i-related hyperglucagonemia and ketone production [65]. Finally, SGLT2i increases plasma ketone levels because of overproduction [66]. Interestingly, the increased production of ketone bodies has the potential to enhance cardiac metabolic efficiency. Studies on human and animal models have proven the role of β-hydroxybutyrate in reversing ventricular remodeling, resulting in improvement of cardiac output and diastolic function [67,68,69]. However, a meta-analysis by Monami et al. analyzed 72 RCTs of T2DM patients in which only nine reported at least one incidence of diabetic ketoacidosis. Therefore, the risk of diabetic ketoacidosis due to the SGLT2i class (MH-OR [95% CI]. 1.14 [0.45–2.88], *p* = 0.78) is negligible [70].

### 8.3. Hypotension

SGLT2i have an osmotic diuretic effect, and this can lead to slight volume depletion. This phenomenon occurs due to the depletion of glucose and thus sodium, commonly leading to symptoms such as decreased blood pressure, orthostatic hypotension, and dizziness [71]. However, a meta-analysis by Liakos et al. compared the depletion of body volume in patients taking CANA or a placebo and concluded that there was no significant difference [72]. Similarly, a study by Baker et al. did not show a significant volume depletion or impact on orthostatic hypotension in patients receiving SGLT2i (*p* > 0.05) [73].

### 8.4. Hypoglycemia

SGLT2i therapy is associated with low risk of hypoglycemia [11,74,75]. The incidence rates of SGLT2i-induced hypoglycemia are similar to current antidiabetic drugs such as MET, glitazones, and DDP-4 inhibitors [71], and they are substantially lower when compared to insulin or SU. It is advised to reduce the dosage of SU when adding SGT2i to the regimen [71].

### 8.5. Lower Limb Amputations

As demonstrated in the CANVAS and CANVAS-Renal (CANVAS-R), CANA was affiliated with a two-fold higher risk of amputations of the toes and metatarsals when compared with placebo (6.3 vs. 3.4/1000 patient-years) [11]. The CANVAS program patients had a mean age of 63 years, and each participant reported a CV condition or peripheral vascular disease at baseline, suggesting that these two factors may impact the final result. It was further investigated in a cohort study by Fralick et al. [76] of three US national databases of patients administered CANA and GLP-1 agonists. The study concluded that the risk of amputations whilst taking CANA was greater in comparison to GLP agonists [76]. Patients with a history of amputation or peripheral vascular disease naturally have the highest absolute risk of amputation [11]. On the other hand, the Empagliflozin Cardiovascular Outcome Event Trial in Type 2 Diabetes Mellitus Patients (EMPA-REG OUTCOME) and the Multicenter Trial to Evaluate the Effect of Dapagliflozin on the Incidence of Cardiovascular Events (DECLARE-TIMI 58) did not confirm the association between SGLT2i and the higher risk of lower limb amputations [74,75]. 

### 8.6. Bone Fractures

The CANVAS arm of the CANVAS program highlighted an increased risk of non-vertebral fractures during treatment with CANA compared with placebo (HR 1.55), in contrast to the CANVAS-R trial (HR 0.86) [11,77]. While CANVAS-R showed an increase in renal impairment, this group had lower risk of fractures than those in the CANVAS study [11,77]. Furthermore, the Evaluation of Canagliflozin’s Effects on Renal and Cardiovascular Outcomes in Participants With Diabetic Nephropathy (CREDENCE) trial reported null result for fractures; therefore, the observation in the CANVAS was most likely incidental [34].

## 9. Cardiovascular Outcomes of SGLT2i

### 9.1. EMPA-REG OUTCOME Trial (2015): Empagliflozin—Cardiovascular Outcomes and Mortality in Diabetic Patients 

The EMPA-REG OUTCOME trial in 2015 was a pioneering randomized double-blind placebo study that evaluated the effect of SGLT2i on CV diseases. The trial was conducted in 590 sites across 42 countries in 7020 patients who matched the following eligibility criteria: age > 18 years, body mass index < 45, estimated glomerular filtration rate (eGFR) > 30 mL/min, established CV disease, HBA1c 7–9% if not on a glucose-lowering agent for the past 12 weeks or HBA1c 7–10% if on a stable glucose-lowering agent for the past 12 weeks. The primary outcomes that were evaluated were 3-point major CV events (MACE): death from CV causes, non-fatal MI, and non-fatal stroke. Secondary outcomes measured were a 4-point MACE: CV death, non-fatal MI, non-fatal stroke, and hospitalization for unstable angina. The median follow-up period was 3.1 years. The study revealed that the 3-point MACE occurred significantly less frequently in the EMPA group (490/4687 (10.5%)) than in the placebo group 282/2333 (12.1%; HR 0.86; 95% CI 0.74–0.99; *p* < 0.001 for noninferiority and *p* = 0.04 for superiority in the treatment group). The 4-MACE secondary outcomes occurred in 599/4687 patients (12.8%) in the EMPA group, and 333/2333 patients (14.3%) in the placebo group (HR 0.89; 95% CI 0.78–1.01; *p* < 0.001 for noninferiority and *p* = 0.08 for superiority). There was no significant difference between the two groups. Moreover, the EMPA group had a significantly reduced risk of death from CV causes (HR 0.62; 95% CI 0.49–0.77; *p* < 0.001), all-cause mortality (HR 0.68; 95% CI 0.57–0.82; *p* < 0.001), and recurrent hospitalizations for HF exacerbations (HR 0.65; 95% CI 0.50–0.85; *p* = 0.002). However, there was no significant difference between the risk of MI or stroke between the treatment and the placebo groups as well as no significant difference between the dose–effect of 10 and 25 mg of EMPA on CV events [75].

The EMPA-REG OUTCOME trial showed clearly that EMPA is superior to placebo in improving glycemic control and reducing CV events including death and HF hospitalizations [75].

### 9.2. CANVAS Program (2017): Canagliflozin—Cardiovascular and Renal Outcomes in Diabetic Patients 

The CANVAS program consisted of integrated trials—CANVAS–Renal (CANVAS-R; n = 5812) designed to assess effects on albuminuria, and CANVAS (n = 4330) for evaluating CV effects of CANA. The CANVAS program was conducted across 667 centers in 30 countries. Eligible patients had a diagnosis of T2DM and a high CV risk, and they were randomly assigned in a 1:1:1 ratio to receive CANA at doses of 100 mg, 300 mg, or placebo. Patients in CANVAS-R were randomly assigned in a 1:1 ratio to receive CANA at an initial dose of 100 mg with an optional increase to 300 mg.

The primary outcome was a 3-point MACE consisting of death from CV causes, non-fatal MI, or non-fatal stroke. Secondary outcomes were death from any cause, death from CV causes, hospitalization for HF, and the progression of albuminuria. 

The primary outcome occurred significantly less frequently in the treatment group (26.9/1000 patient-years) than in the placebo group (31.5/1000 patient-years); HR 0.86; 95% CI 0.75–0.97; *p* < 0.001 for noninferiority; *p* = 0.02 for superiority. Risk of hospitalization due to HF was also significantly lower in the CANA group (5.50 events per 1000 patient-years) than in the placebo (8.68 events per 1000 patients); HR 0.67; 95% CI 0.52–0.87.

Albuminuria progressed less frequently in the treatment group (89.4 events per 1000 patients) than in the placebo group (128.7 events per 1000 patients); HR 0.73; 95% CI 0.67–0.79. The risk of amputation was higher with CANA (6.3 per 1000 patient-years) compared to placebo (3.4 per 1000 patient-years); HR 1.97; 95% CI 1.41–2.75, especially in patients with a prior amputation or peripheral arterial disease. This was a novel finding in this class of drugs. Moreover, the all-fracture event rate was higher with CANA (15.4 per 1000 patient-years) compared to placebo (11.9 per 1000 patient-years); HR 1.26; 95% CI 1.04–1.52. Similar findings were made with low trauma fractures, with a higher event rate in the CANA group (11.6 fracture per 1000 patient-years) compared to placebo (9.2 fracture per 1000 patient-years); HR 1.23; 95% CI 0.99–1.52. 

The CANVAS program revealed a significantly reduced risk of CV events in T2DM patients treated with CANA. Moreover, CANA is potentially beneficial regarding the progression of albuminuria. However, the rate of amputations and bone fractures was greater in the treatment group [11,77].

### 9.3. DECLARE-TIMI 58 Trial (2018): Dapagliflozin—Cardiovascular Safety Profile in Diabetic Patients

The Dapagliflozin Effect on Cardiovascular Events–Thrombolysis in Myocardial Infarction 58 (DECLARE–TIMI 58) trial was designed to establish the effect of DAPA on CV and renal outcomes in diabetic patients.

The trial was conducted in 882 sites in 33 countries. A total of 17,160 enrolled patients were followed for a median of 4.2 years, and the study uniquely included 10,186 individuals without diagnosed CV disease. Eligible patients were given 10 mg of DAPA or placebo in a 1:1 ratio. The primary outcome was a 3-point MACE defined as death due to CV reasons, MI, or ischemic stroke. Two primary efficacy outcomes were MACE and a composite of CV death or hospitalization for HF. Two secondary efficacy outcomes were renal composite outcome (≥40% reduction in eGFR, new ESRD, or death from renal or CV causes) and all-cause mortality.

The trial revealed DAPA to be noninferior to placebo but failed to show any significantly lower rate of MACE in the study group. MACE occurred in 8.8% (756/8582) of the DAPA group and 9.4% (803/8578) of the placebo group; HR 0.93; 95% CI 0.84–1.03; *p* < 0.001 for noninferiority; *p* = 0.17 for superiority. However, CV death or hospitalizations for HF occurred less frequently in the treatment group (4.9%) compared to placebo (5.8%); HR 0.83; 95% CI 0.73–0.95; *p* = 0.005. This finding was consistent across multiple subgroups, including patients with and without a history of HF. The renal events occurred in 4.3% of patients in the DAPA group compared to 5.6% in the placebo group; HR 0.93; 95% CI 0.82–1.04. 

In contrast to the previous trials, the rates of adverse events such as amputations, fractures, hypovolemia, and hypersensitivity reactions were similar in the treatment and placebo groups. However, diabetic ketoacidosis and genital infections were more frequent in the DAPA group (0.9% vs. 0.1%; *p* < 0.001).

Although the DECLARE-TIMI 58 trial has not shown DAPA to be superior concerning CV deaths or all-cause mortality, it highlighted the ability of DAPA to prevent hospitalizations for HF and the progression of renal failure in patients without established CV disease [74]. 

### 9.4. CREDENCE Trial (2019): Canagliflozin—Renal Outcomes in Diabetes and Nephropathy 

The Evaluation of Canagliflozin’s Effects on Renal and Cardiovascular Outcomes in Participants With Diabetic Nephropathy (CREDENCE) trial was a double-blind, randomized event-driven trial designed to further evaluate the effect of CANA on renal function in T2DM patients with pre-existing kidney disease in a setting of therapy with angiotensin-converting-enzyme inhibitors or angiotensin receptor blockers. The primary outcome was a composite of ESRD, doubling of the baseline serum creatinine level ≥30 days or death due to renal or CV disease. Secondary outcomes were the composite of CV death, hospitalization due to HF or unstable angina, MI, stroke, ESRD, doubling of the baseline serum creatinine, dialysis, kidney transplantation, and finally renal or CV death. The CREDENCE study group was particularly noteworthy due to the majority of patients (60%) with an eGFR of 30 ≤ 60 mL/min/1.73 m^2^, while previous SGLT2i trials implemented a broad inclusion criterion of eGFR ≥ 30 mL/min/1.73 m^2^. 

A total of 4401 eligible patients were randomly assigned in a 1:1 ratio to receive either 100 mg of CANA or a placebo. Patients were stratified as per eGFR results. They were intended to be followed up for 5 years. However, the trial was terminated at 2.62 years due to interim analysis findings that revealed significant benefits in the treatment group. 

The primary outcome occurred in significantly lower rates in the CANA group (43.2 per 1000 patient-years) than in the placebo group (61.2 per 1000 patient-years); HR 0.70; 95% CI 0.59–0.82; *p* = 0.00001. The findings were consistent across the subgroups (sorted according to blood glucose levels, weight, and blood pressure) and for the renal components, including ESRD; HR 0.68; 95% CI 0.54–0.86; *p* = 0.002. 

Secondary outcomes were also significantly lower in the treatment group compared to the placebo group, including composite CV outcomes: CV death or hospitalization for HF (HR 0.69; 95% CI 0.57–0.83; *p* < 0.001), CV death, MI, or stroke (HR 0.80; 95% CI 0.67–0.95; *p* = 0.01), and hospitalization for HF (HR 0.61; 95% CI 0.47–0.80; *p* < 0.001). The results of this trial in terms of CV death, MI, stroke, and reduction in the number of hospitalizations for HF reinforced the outcomes of the EMPA-REG and CANVAS trials. Moreover, it further showed the superiority of CANA in comparison to placebo, irrespective of the glucose levels, which opened up the possibility of exploring the effect of CANA and other SGLT2i in non-diabetic patients. The rates of amputations and fractures observed in the CANVAS program were not seen in the CREDENCE trial [34].

### 9.5. DAPA-HF Trial (2019): Dapagliflozin—Impact on Patients with Heart Failure and Reduced Ejection Fraction Regardless of Diabetes

Prior to the Dapagliflozin and Prevention of Adverse Outcomes in Heart Failure (DAPA-HF) trial, trials were mainly investigating the effect of SGLT2i on diabetic patients without HF at baseline. In other words, they only showed the efficacy of SGLT2i in preventing the incidence of HF, which are results that are not relevant to patients with established HF. The DAPA-HF trial took a step in a different direction by evaluating the effects of SGLT2i on patients with pre-existing HF and HFrEF with or without pre-existing T2DM. 

A total of 4744 eligible patients from 410 centers in 20 countries were assigned to receive either 10 mg of DAPA or an equivalent placebo dose. 

The primary outcome was a composite of worsening HF (measured by acute hospitalization or being treated with an intravenous HF treatment after an urgent visit due to HF) or death from CV causes. The key secondary outcome was a composite of hospitalization due to HF or CV death. The other secondary outcomes included the total number of hospitalizations for HF, CV, and all-cause mortality, and renal outcomes.

The results showed a superiority of DAPA compared to placebo regarding the primary outcome. The event rate was 16.3% and 21.2% in the DAPA and placebo groups, respectively; HR 0.74; 95% CI 0.65–0.85; *p* < 0.001). Hospitalizations due to HF occurred less frequently in the DAPA group (9.7%) in comparison to placebo (13.4%); HR 0.70; 95% CI 0.59–0.83). CV death occurred at a rate of 9.6% in the treatment group and 11% in the placebo group; HR 0.82; 95% CI 0.69–0.98. The superiority of DAPA was also observed in the subgroups, including patients without T2DM. 

Moreover, the secondary outcomes also showed the superiority of DAPA over placebo. The number of hospitalizations due to HF or CV death was significantly lower in the treatment group; HR 0.75; 95% CI 0.65–0.85; *p* < 0.001. The all-cause mortality rate was 11.6% in the DAPA group and 13.9% in the placebo group; HR 0.83; 95% CI 0.71–0.97. 

The DAPA-HF trial was a crucial study for HF patients and demonstrated the similar impact of DAPA on this population irrespective of T2DM [78].

### 9.6. EMPEROR-Reduced Trial (2020): Empagliflozin—Cardiovascular and Renal Outcomes in Heart Failure Patients 

The Empagliflozin Outcome Trial in Patients with Chronic Heart Failure and a Reduced Ejection Fraction (EMPEROR-Reduced) trial focused exclusively on HF patients with HFrEF with or without T2DM who were already taking guideline-directed medical therapy for HF. 

A total of 3730 eligible patients diagnosed with HFrEF received 10 mg of EMPA daily or an equivalent placebo. They were followed up for a median of 16 months.

The primary outcome was a composite of CV death or hospitalization for HF (analyzed as the time to the first event). The secondary outcomes were the event rates of all adjudicated hospitalizations for HF, including first and recurrent events and decline in the eGFR.

EMPEROR-Reduced showed a superiority of EMPA over the placebo in terms of the primary outcome, which occurred at a rate of 19.4% in the treatment group and 24.7% in the placebo group; HR 0.75; 95% CI 0.65–0.86; *p* < 0.001. The superiority of EMPA was also demonstrated in all subgroups, including patients with and without T2DM. The same correlation regarding secondary outcomes, e.g., the total number of hospitalizations, was higher in the placebo group than in the treatment group; HR 0.70; 95% CI 0.58–0.85; *p* < 0.001. Moreover, patients treated with EMPA had a lower risk of serious renal outcomes. The eGFR declined slower in the treatment group, regardless of T2DM. Nevertheless, uncomplicated genital tract infections were reported more frequently in the study group. The adverse effects of hypoglycemia, limb amputations, and fractures were similar in both groups.

As in the DAPA-HF trial, the EMPEROR-Reduced trial demonstrated that patients with HF benefited from EMPA therapy, regardless of the presence or absence of T2DM [32].

### 9.7. SOLOIST-WHF Trial (2020): Sotagliflozin—Impact on Patients with Decompensated Heart Failure and Diabetes

The recently published Effect of Sotagliflozin on Cardiovascular Events in Patients With Type 2 Diabetes Post Worsening Heart Failure (SOLOIST-WHF) trial intended to examine the effects of SGLT2i in patients with T2DM and decompensated HF. A total of 1222 patients from 32 countries were enrolled in this randomized, double-blind, and placebo-controlled study. A total of 608 participants in the study group received 200 mg of SOTA once daily either before or within three days after hospital discharge, and the remaining received placebo. The trial ended much earlier than anticipated due to decreased funding, with a median follow-up of 9 months. The primary endpoint was the total number of deaths due to CV events, hospitalizations, and urgent visits for HF. 

A total of 600 primary endpoint events occurred in the SOTA group (245) and in the placebo group (355). The rate of primary endpoint events (number of events/100 patient-years) was lower in the SOTA group (51.0) than in the placebo group (76.3); HR 0.67; 95% CI 0.52–0.85; *p* < 0.001. 

The trial showed that SOTA had a significant impact on reducing the total number of CV deaths, hospitalizations, and urgent visits for HF in patients with T2DM and decompensated HF. Interestingly, the placebo group experienced a higher rate of primary endpoints in the first 90 days after randomization; therefore, the early initiation of SGLT2i may be essential in preventing adverse events [79].

### 9.8. EMPEROR-Preserved Trial (2021): Empagliflozin—Cardiovascular Outcomes in Heart Failure Patients 

The Empagliflozin Outcome Trial in Patients with Chronic Heart Failure and Preserved Ejection Fraction (EMPEROR-Preserved) trial enrolled 5988 patients with HF and EF > 40% (HFpEF) with and without T2DM who are randomized to receive 10 mg of EMPA daily or placebo as an add-on to the standard treatment. The results of this double-blind trial were presented at the 2021 ESC Congress and showed that over a median observation time of 26.2 months, the EMPA group demonstrated a decreased composite risk of CV death or hospitalization for HF (415 of 2997, 13.8%) than the placebo group (511 of 2991, 17.1%), (HR 0.79; 95% CI 0.69–0.9). Fewer EMPA patients died from CV causes (7.3%; compared to placebo 8.2%; HR 0.91; 95% CI 0.76–1.09), and EMPA patients were less frequently hospitalized (8.6%; compared to placebo 11.8%; HR 0.71; 95% CI 0.60–0.83). This was consistent over patients with or without diabetes. The EMPA group reported more frequent uncomplicated genital (2.2%; compared to placebo 0.7%) and urinary tract infections (9.9%; compared to placebo 8.1%) as well as hypotension, but in general, EMPA patients reported fewer (47.9%) serious adverse effects than placebo (51.6%).

From the results of this trial, EMPA appears to be a promising SGLT2i—and the first agent on the market—for reducing the composite risk of CV death or HF hospitalization in HFpEF patients, with or without diabetes comorbidity [80].

A summary of the key information regarding clinical trials with SGLT2i is provided in Table 3.

## 10. Summary

SGLT2i are effective antidiabetic therapies in T2DM patients that are associated with better glycemic control and reductions in body mass and blood pressure. Recent large, multicenter trials have proven that SGLT2i also improve CV and renal outcomes, including reduced CV mortality and fewer hospitalizations for HF. It is particularly noteworthy that these advantages extend to HF patients without T2DM; thus, it is highly unlikely that SGLT2i’s beneficial effects are related to improved glycemic control alone. Reduced preload and afterload, improved vascular function, and changes in tissue sodium and calcium handling are thought to play a role.

Thus arises the fundamental question—are the effects of SGLT2i drug- or class-specific? They have similar chemical structures, mechanisms of action, and pharmacological effects, and they undeniably improve glycemic control in T2DM patients. A meta-analysis by Zannad et al. of two large HFrEF trials, DAPA-HF and EMPEROR-Reduced, has shown no heterogeneity in efficacy outcomes [82]. Both trials established SGLT2i—independent of diabetes status and glycemic effects—as effective and well-tolerated therapies reducing CV mortality, number of hospitalizations for HF, and improving quality of life in HFrEF patients. Verma et al. have even stipulated that SGLT2i should be considered a crucial component of goal-directed medical therapy in HFrEF [83]. On the other hand, a meta-analysis by Zelniker et al. of the CANVAS, DECLARE-TIMI 58, and EMPA-REG trials showed the following overall patterns. Firstly, SGLT2i clearly reduces the risk of MACE but only in patients with existing atherosclerotic cardiovascular disease (ACVD). Furthermore, only EMPA reduces all-cause and CV mortality in those patients. Surprisingly, the class as a whole does not seem to have any impact on the risk of stroke. However, it is explicitly important that all SGLT2i reduce the risk of hospitalization for HF, regardless of prior ACVD or HF. Lastly, all SGLT2i increase the risk of mycotic genital infections (usually easily managed) as well as diabetic ketoacidosis, although the rates were very low. Only CANA increases the risk of amputations and fractures [84]. McGuire et al. performed a meta-analysis of six trials (EMPA-REG OUTCOME, CANVAS and CANVAS-R, DECLARE-TIMI 58, CREDENCE, and VERTIS CV on ERTU) [85], including a total of 46,969 unique T2DM patients; 66.2% had ACVD. SGLT2i were associated with a reduced risk of MACE (HR 0.90; 95% CI 0.85–0.95), and hospitalization for HF or CV death (HR 0.78; 95% CI, 0.73–0.84), with no significant heterogeneity of associations with outcome. Reduced risk of hospitalization due to HF was consistent across the trials (HR 0.68; 95% CI 0.61–0.76), whereas significant heterogeneity of associations with outcome was observed for CV death (HR 0.85; 95% CI 0.78–0.93). Interestingly, the presence or absence of ACVD did not modify the association with outcomes for major adverse CV events [86]. According to Giugliano et al., there is no class effect of SGLT2i regarding MACE, as the DECLARE-TMI 58 trial failed to prove DAPA’s superiority in terms of reducing risk of CV death, MI, or stroke. However, the class effect was evident for HF and diabetic kidney disease. In the EMPA-REG OUTCOME, CANVAS, DECLARE-TIMI 58, and CREDENCE trials, all SGLT2i significantly reduced the risk of hospitalization for HF and progression of renal failure [87]. SGLT2i seem to play a particularly important role in reducing the risk of HF hospitalizations. As a result of these widely consistent findings across numerous trials, McMurray and Packer proposed new guidelines of HF management in 2019, proposing that the addition of a new drug class is more beneficial than the up-titration of current drugs (the starting and target doses are identical for SGLT2i, however). Furthermore, sequencing of drug classes may improve safety and tolerability. SGLT2i can minimize the risk of hyperkalemia with the use of mineralocorticoid receptor antagonists. Much of the benefits of foundational treatments are seen within 30 days after initiation; therefore, McMurray and Packer’s algorithm assumes achievement of therapy in three steps with all four drug classes within 4 weeks. Courageously, step 1 is the simultaneous initiation of treatment with β-blocker and SGLT2i, which may diminish the short-term risk of HF worsening due to the latter agent. As more and more trials convincingly point to the beneficial effects of SGLT2i, the newest ESC guidelines are advising the addition of DAPA and EMPA as first-line agents for patients with HFrEF in combination with other recommended first-line therapy regardless of T2DM comorbidity. The proposed algorithms are presented in Figure 4 [88].

SGLT2i are efficient in different stages of HF, from pre-HF to advanced HF, as well as in acute and chronic phases of the disease. Severino et al. assume that HF, similar to a cancer, should be managed in order to evade the progressive involvement and failure of systemic organs, and therefore propose that SGLT2i should be administered as soon as possible [89] in order to take advantage of their cardio- and nephroprotective benefits [39]. The diuretic and natriuretic properties of SGLT2i convey additional benefits by reducing congestion and allowing for the reduction in loop diuretic doses, which are typically very high. CANA, DAPA, EMPA, ERTU, and SOTA are recommended in T2DM patients at risk of CV events in order to reduce hospitalizations due to HF, MACE, ESRD, as well as CV death. Most excitingly, the recent results of the EMPEROR-Preserved trial give us hope that there is a light at the end of the tunnel for the management of patients with HFpEF, for whom previous pharmaceutical interventions have been largely neutral [6,80].

## Figures and Tables

**Figure 1 jcm-11-01470-f001:**
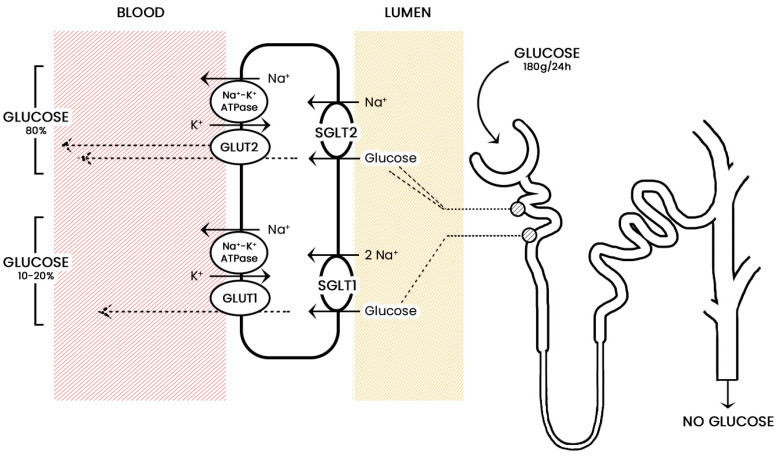
The mechanism of action and location of sodium-glucose co-transporter (SGLT) proteins. SGLT2 are found in the proximal (S1) segments, whilst the SGLT1 are found in the S2 and S3 portions of the proximal tubule. Figure reprinted with permission from ref. [10]. 2016, Springer Nature. Abbreviations: GLUT1—glucose transporter 1, GLUT2—glucose transporter 2, Na⁺/K⁺-ATPase—sodium–potassiumadenosine triphosphatase, SGLT1—sodium-glucose co-transporter 1, SGLT2—sodium-glucose co-transporter 2.

**Figure 2 jcm-11-01470-f002:**
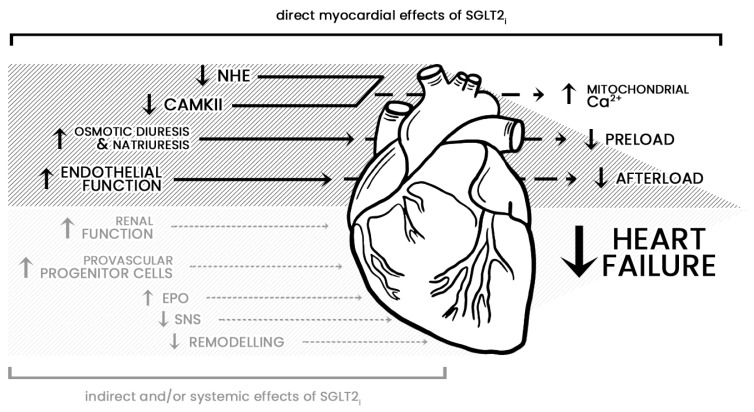
The mechanism of action of sodium-glucose co-transporter-2 inhibitors on the cardiovascular system. Figure reprinted with permission from ref. [38]. 2020 Dr. Gary Lopaschuk. Abbreviations: NHE—sodium-hydrogen exchange, CAMKII—calmodulin-dependent protein kinase II, EPO—erythropoietin, SNS—sympathetic nervous system, Ca^2+^—calcium.

**Figure 3 jcm-11-01470-f003:**
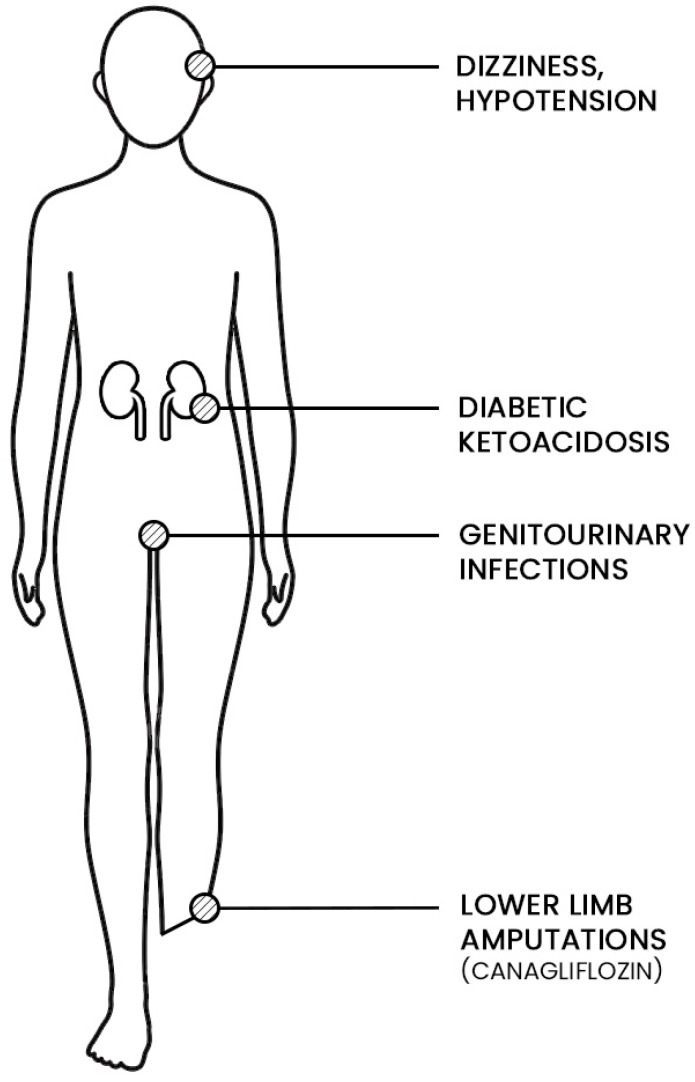
Major adverse effects of SGLT2i.

**Figure 4 jcm-11-01470-f004:**
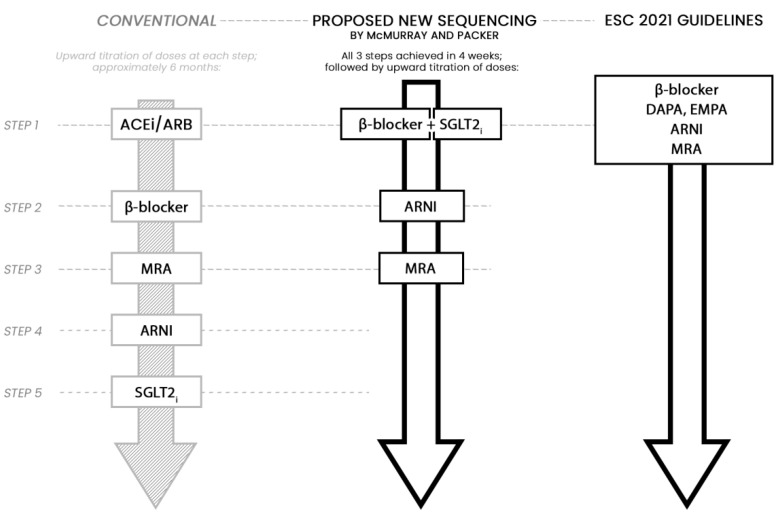
Proposal of a new algorithm for sequencing of foundational treatments in heart failure. Figure based on [6,88]. (Source: American Heart Association, Inc.; European Society of Cardiology). Abbreviations: ACEi—angiotensin-converting enzymes inhibitor, ARB—angiotensin receptor blockers, ARNI—angiotensin receptor neprilysin inhibitor, DAPA—dapagliflozin, EMPA—empagliflozin, ESC—European Society of Cardiology, MRA—mineralocorticoid receptor antagonist, SGLT2i—sodium-glucose co-transporter 2 inhibitor.

**Table 1 jcm-11-01470-t001:** Dosage of SGLT2 inhibitors. Table on the basis of summaries of product characteristics.

Generic Name	Brand Name	Starting Dose (mg) Once Daily	Maintenance Dose (mg) Once Daily
Canagliflozin	Invokana	100	100–300
Dapagliflozin	Farxiga Forxiga	10	10
Empagliflozin	Jardiance	10	10–25
Ertugliflozin	Steglatro	5	5–15

**Table 2 jcm-11-01470-t002:** SGLT2 inhibitors corresponding with renal function. Table on the basis of summaries of product characteristics.

	Dose (mg)	eGFR Cutoffs (mL/min/1.73 m^2^) for Patients with T2DM	Recommendations for Patients with T2DM	eGFR Cutoffs (mL/min/1.73 m^2^) for Patients with HF	Recommendations for Patients with HF
Canagliflozin	100–300	60	Max 300 mg	-	-
		30–59	Max 100 mg		
		<30	Max 100 mg for patients already taking CANA, otherwise should not be initiated		
Dapagliflozin	10	≥45	10 mg	≥25	10 mg
		<45	Additional glucose-lowering treatment may be needed	<25	10 mg for patients already taking DAPA, otherwise should not be initiated
		<25	10 mg for patients already taking DAPA, otherwise contraindicated		
Empagliflozin	10–25	≥60	Max 25 mg	≥20	10 mg
		45–59	Max 10 mg, only if eGFR before treatment ≥60	<20	Contraindicated
		<45	Contraindicated		
Ertugliflozin	5–15	<60	Should not be initiated	-	-
		<45	Contraindicated		

Abbreviation: eGFR—estimated glomerular filtration rate. CANA, DAPA, and EMPA can be used as monotherapy or in combination with other antidiabetic drugs such as MET and insulin. Currently, these are only approved for patients with T2DM; more trials are needed for T1DM [43].

**Table 3 jcm-11-01470-t003:** Summary of clinical trials with SGLT2i—cardiovascular outcomes.

	EMPA-REG OUTCOME [75]	CANVAS Program [77]	DECLARE-TIMI-58 [36]	CREDENCE [77]	DAPA-HF [78]	EMPEROR- Reduced [81]	SOLOIST- WHF [79]	EMPEROR- Preserved [80]
Year	2015	2017	2018	2019	2019	2020	2020	2021
Randomized	yes	yes	yes	yes	yes	yes	yes	yes
Double-blind	yes	yes	yes	yes	yes	yes	yes	yes
Placebo-controlled	yes	yes	yes	yes	yes	yes	yes	yes
SGLT2i	EMPA	CANA	DAPA	CANA	DAPA	EMPA	SOTA	EMPA
Study drug doses	10, 25 mg	100, 300 mg	10 mg	100 mg	10 mg	10 mg	200, 400 mg (uptitrated if tolerated)	10 mg
Number of randomized patients	7.020	10.142	17.160	4.401	4.744	3.730	1.222	5.988
Median observation time	3.1 years	2.4 years	4.2 years	2.6 years	18.2 months	16 months	9 months	26.2 months
Key inclusion criteria								
Age	≥18 y.o	≥30 y.o and a history of symptomatic ASCVD or ≥50 y.o with ≥2 CV risk factors	≥40 y.o and a history of ASCVD or ≥55 y.o (men) or ≥60 y.o (women) without known ASCVD, but ≥1 CV risk factor	≥30 y.o	≥18 y.o	≥18 y.o	18–85 y.o	≥18 y.o
T2DM	yes	yes	yes	yes	yes/no	yes/no	yes	yes/no
HbA1C	7–9% (if no GLT for ≥12 weeks before randomization) 7–10% (if GLT for ≥12 weeks before randomization)	7–10.5%	6.5–12%	6.5–12%	-	-	≥6.5%	-
ASCVD	CAD MI (>2 months ago) Stroke (>2 months ago) PAD	as above	IHD ICVD PAD	-	-	-	-	-
Heart failure	-	-	-	-	LVEF ≤ 40% NYHA class II-IV NT-proBNP: ≥900 pg/mL (if AF/AFL) ≥400 pg/mL (if hospitalization for HF within 1 year) ≥600 (in neither of the above)	LVEF ≤ 40% NYHA class II-IV NT-proBNP: LVEF ≤30%: ≥600 pg/mL (≥1200 pg/mL, if AF) LVEF 31–35%: ≥1000 pg/mL (≥2000 pg/mL, if AF) LVEF 36–40%: ≥2500 pg/mL (≥5000 pg/mL, if AF)	Acute HF requiring IV diuretic therapy (clinically stable at randomization)	LVEF > 40% NYHA class II-IV NT-proBNP: >300 pg/mL >900 pg/mL (if AF)
eGFR [mL/min/1.73 m^2^]	≥30	≥30	≥60 (creatinine clearance)	30–<90 (30–60 in 60% of study population) + albuminuria—UACR: >300–5000 mg/g	≥30	≥20	≥30	≥20
Other	-	-	-	Stable maximum tolerated daily dose of ACEi or ARB	GDMT for HF for and cardiac device therapy as indicated	GDMT for HF for and cardiac device therapy as indicated	-	-
Key exclusion criteria								
Cardiovascular	Stroke/TIA within 2 months Planned cardiac surgery or PCI in next 3 months	-	Acute CV events	Current or history of NYHA class IV	Symptomatic hypotension or SBP <95 mmHg	Acute HF HTX or LVAD Cardiomyopathy other than DCM or HCM without LVOTO Symptomatic hypotension or SBP <100 mmHg	End-stage HF Recent ACS, PCI, CABG, stroke	MI, CABG, or other major CV surgery, stroke, TIA within last 90 days HTX or LVAD Cardiomyopathy (with some exclusions) Severe VHD Acute HF ICD (within 3 months) or CRT
Other	Cancer within 5 years BMI > 45 Hemolysis or unstable RBCs ALT, AST, or ALP > 3 × ULN	History of diabetic ketoacidosis History of severe hypoglycemia within 6 months T1DM or DM secondary to pancreatitis or pancreatomy Pancreas or β-cell transplantation	Poor medication adherence Previous SGLT2i use Steroid use BP > 180/100 mmHg	T1DM History of diabetic ketoacidosis Renal disease requiring immunosuppression Significant liver disease Potassium level >5.5 mmol/L	T1DM Recent SGLT2i use	BMI ≥ 45	-	Significant chronic pulmonary disease or primary pulmonary arterial hypertension Hemoglobin < 9 g/dL Severe liver disease
Results								
Primary outcome	CV death, MI, stroke	CV death, MI, stroke	CV death, MI, stroke	ESKD, doubling of serum creatinine, death from renal or CV causes	Worsening HF (hospitalization or urgent visit resulting in IV therapy), CV death	CV death, HF hospitalization	CV death, HF hospitalizations and urgent visits (first and subsequent)	CV death, hospitalization for HF
	10.5% (EMPA) vs. 12.1% (placebo); (HR 0.86; 95% CI 0.74–0.99)	26.9/1000 patient-years (CANA) vs. 31.5/1000 patient-years (placebo); (HR 0.86; 95% CI 0.75–0.97)	8.8% (DAPA) vs. 9.4% (placebo); (HR 0.93; 95% CI 0.84–1.03)	43.2/1000 patient-years (CANA) vs. 61.2/1000 patient-years (placebo); (HR 0.70; 95% CI 0.59–0.82)	16.3% (DAPA) vs. 21.2% (placebo); (HR 0.74; 95% CI 0.65–0.85)	19.4% (EMPA) vs. 24.7% (placebo); (HR 0.75; 95% CI 0.65–0.86)	51.0/100 patient-years (SOTA) vs. 76.3/100 patient-years (placebo); (HR 0.67; 95% CI 0.52–0.85)	13.8% (EMPA) vs. 17.1% (placebo); (HR 0.79; 95% CI 0.69–0.9)
Death from CV causes	3.7% (EMPA) vs. 5.9% (placebo); (HR 0.62; 95% CI 0.49–0.77)	11.6/1000 patient-years (CANA) vs. 12.8/1000 patient-years (placebo); (HR 0.87; 95% CI 0.72–1.06)	2.9% (DAPA) vs. 2.9% (placebo); (HR 0.98; 95% CI 0.82–1.17)	19.0/1000 patient-years (CANA) vs. 24.4/1000 patient-years (placebo); (HR 0.78; 95% CI 0.61–1.00)	9.6% (DAPA) vs. 11.5% (placebo); (HR 0.82; 95% CI 0.69–0.98)	10.0% (EMPA) vs. 10.8% (placebo); (HR 0.92; 95% CI 0.75–1.12)	51.0/100 patient-years (SOTA) vs. 58.0/100 patient-years (placebo); (HR 0.84; 95% CI 0.58–1.22)	7.3% (EMPA) vs. 8.2% (placebo); (HR 0.91; 95% CI 0.76–1.09)
Death from any cause	5.7% (EMPA) vs. 8.3% (placebo); (HR 0.68; 95% CI 0.57–0.82)	17.3/1000 patient-years (CANA) vs. 19.5/1000 patient-years (placebo); (HR 0.87; 95% CI 0.74–1.01)	6.2% (DAPA) vs. 6.6% (placebo); (HR 0.93; 95% CI 0.82–1.04)	29.0/1000 patient-years (CANA) vs. 35.0/1000 patient-years (placebo); (HR 0.83; 95% CI 0.68–1.02)	11.6% (DAPA) vs. 13.9% (placebo); (HR 0.83; 95% CI 0.71–0.97)	13.4% (EMPA) vs. 14.2% (placebo); (HR 0.92; 95% CI 0.77–1.10)	65.0/100 patient-years (SOTA) vs. 76.0/100 patient-years (placebo); (HR 0.82; 95% CI 0.59–1.14)	14.1% (EMPA) vs. 14.3% (placebo); (HR 1.00; 95% CI 0.87–1.15)
HF hospitalization	2.7% (EMPA) vs. 4.3% (placebo); (HR 0.65; 95% CI 0.50–0.85)	5.5/1000 patient-years (CANA) vs. 8.7/1000 patient-years (placebo); (HR 0.67; 95% CI 0.52–0.87)	2.5% (DAPA) vs. 3.3% (placebo); (HR 0.73; 95% CI 0.61–0.88)	15.7/1000 patient-years (CANA) vs. 25.3/1000 patient-years (placebo); (HR 0.61; 95% CI 0.47–0.80)	9.7% (DAPA) vs. 13.4% (placebo); (HR 0.70; 95% CI 0.59–0.83)	13.2% (EMPA) vs. 18.3% (placebo); (HR 0.69; 95% CI 0.59–0.81)	194.0/100 patient-years (SOTA) vs. 297.0/100 patient-years (placebo); (HR 0.64; 95% CI 0.49–0.83) (total number of HF hospitalizations and urgent visits)	8.6% (EMPA) vs. 11.8% (placebo); (HR 0.71; 95% CI 0.60–0.83)
Safety								
Serious AE	38.2% (EMPA) vs. 42.3% (placebo)	104.3/1000 patient-years (CANA) vs. 120/1000 patient-years (placebo)	34.1% (DAPA) vs. 36.2% (placebo)	145.2/1000 patient-years (CANA) vs. 164.4/1000 patient-years (placebo)	37.8% (DAPA) vs. 42.0% (placebo)	41.4% (EMPA) vs. 48.1% (placebo)	3.0% (SOTA) vs. 2.8% (placebo (only events leading to study drug discontinuation)	47.9% (EMPA) vs. 51.6% (placebo)
Genital infections	6.4% (EMPA) vs. 1.8% (placebo)	34.9/1000 patient-years (CANA) vs. 10.8/1000 patient-years (placebo) (men) 68.8/1000 patient-years (CANA) vs. 17.5/1000 patient-years (placebo) (women)	0.9% (DAPA) vs. 0.1% (placebo)	8.4/1000 patient-years (CANA) vs. 0.9/1000 patient-years (placebo) (men) 12.6/1000 patient-years (CANA) vs. 6.1/1000 patient-years (placebo) (women)	-	1.7% (EMPA) vs. 0.6% (placebo)	-	2.2% (EMPA) vs. 0.7% (placebo)
Urinary tract infections	18.0% (EMPA) vs. 18.1% (placebo)	40.0/1000 patient-years (CANA) vs. 37.0/1000 patient-years (placebo)	1.5% (DAPA) vs. 1.6% (placebo)	48.3/1000 patient-years (CANA) vs. 45.1/1000 patient-years (placebo)	0.5% (DAPA) vs. 0.7% (placebo)	4.9% (EMPA) vs. 4.5% (placebo)	4.8% (SOTA) vs. 5.1% (placebo)	9.9% (EMPA) vs. 8.1% (placebo)
Hypoglycemia	27.8% (EMPA) vs. 27.9% (placebo)	50.0/1000 patient-years (CANA) vs. 46.4/1000 patient-years (placebo)	0.7% (DAPA) vs. 1.0% (placebo)	44.3/1000 patient-years (CANA) vs. 48.9/1000 patient-years (placebo)	0.2% (DAPA) vs. 0.2% (placebo)	1.4% (EMPA) vs. 1.5% (placebo)	1.5% (SOTA) vs. 0.3% (placebo)	2.4% (EMPA) vs. 2.6% (placebo)
DKA	0.1% (EMPA) vs. <0.1% (placebo)	0.6/1000 patient-years (CANA) vs. 0.3/1000 patient-years (placebo)	0.3% (DAPA) vs. 0.1% (placebo)	2.2/1000 patient-years (CANA) vs. 0.2/1000 patient-years (placebo)	0.1% (DAPA) vs. 0.0% (placebo)	0% (EMPA) vs. 0% (placebo)	0.3% (SOTA) vs. 0.7% (placebo)	0.1% (EMPA) vs. 0.2% (placebo)
Amputations	-	6.3/1000 patient-years (CANA) vs. 3.4/1000 patient-years (placebo)	1.4% (DAPA) vs. 1.3% (placebo)	12.3/1000 patient-years (CANA) vs. 11.2/1000 patient-years (placebo)	0.5% (DAPA) vs. 0.5% (placebo)	0.7% (EMPA) vs. 0.5% (placebo)	-	0.5% (EMPA) vs. 0.8% (placebo)
Bottom line								
Summary	Lower rate of the primary composite CV outcome, death from CV and any cause and hospitalizations for HF in patients receiving EMPA in addition to standard care	Lower rate of the primary composite CV outcome and hospitalizations for HF in patients receiving CANA	Noninferior but not superior regarding primary composite CV outcome; however, DAPA reduces risk of hospitalizations for HF	Lower rate of the primary composite CV outcome, death from CV causes and hospitalizations for HF in patients receiving CANA	Lower rate of the primary composite CV outcome, death from CV and any cause and hospitalizations for HF in patients receiving DAPA regardless of presence or absence of T2DM	Lower rate of the primary composite CV outcome, and hospitalizations for HF in patients receiving EMPA regardless of presence or absence of T2DM	Lower rate of the primary composite CV outcome, HF hospitalization and urgent visits in patients receiving SOTA	Lower rate of the primary composite CV outcome, and hospitalizations for HF in patients receiving EMPA regardless of presence or absence of T2DM
Additionalconsiderations	Reduction in CV events apparent after only 3 months The reduction in the primary outcome resulted mainly from a reduction in CV death, not MI or stroke	Decreased progression of albuminuria Increased risk of amputations	Unclear risk of selection bias: only 17,160 randomized patients out of 25,698 in run-in phase	Assessment of CANA in the renal failure population	<5% Black patients Underrepresented old patients with multiple comorbidities	10 mg/d dose based on the lack of difference between 10 and 25 mg/d in the EMPA-REG trial EMPEROR-Reduced and DAPA-HF showed no heterogeneity in efficacy outcomes, which strongly suggests a class effect of SGLT2i	No LVEF restriction Primary outcome consistent across subgroups (LVEF < 50%: HR 0.72 (95% CI 0.56–0.94) and LVEF ≥ 50%: HR 0.48 (95% CI 0.27–0.86)) Prematurely discontinued due to financial reasons	First medication reducing the combined risk of CV death or hospitalization for HF in the HFpEF population

Abbreviations: ACEi—angiotensin-converting enzyme inhibitor, ACS—acute coronary syndrome, AF—atrial fibrillation, AFL—atrial flutter, ALT—alanine transaminase, ALP—alkaline phosphatase, ARB—angiotensin receptor blocker, ASCVD—atherosclerotic cardiovascular disease, AST—aspartate transaminase, BMI—body mass index, BP—blood pressure, CAD—coronary artery disease, CABG—coronary artery bypass grafting, CANA—canagliflozin, CRT—cardiac resynchronization therapy, CV—cardiovascular, DAPA—dapagliflozin, DCM—dilated cardiomyopathy, DM—diabetes mellitus, eGFR—estimated glomerular filtration rate, EMPA—empagliflozin, ESKD—end-stage kidney disease, GDMT—guideline-directed medical therapy, GLT—glucose-lowering therapy, HbA1C—glycated hemoglobin, HCM—hypertrophic cardiomyopathy, HF—heart failure, HFpEF—heart failure with preserved ejection fraction, HR—hazard ratio, HTX—heart transplantation, ICD—implantable cardioverter defibrillator, ICVD—ischemic cerebrovascular disease, IHD—ischemic heart disease, IV—intravenous, LVAD—left ventricular assist device, LVEF—left ventricular ejection fraction, LVOTO—left ventricular outflow tract obstruction, MI—myocardial infarction, NT-proBNP—N-terminal pro-brain natriuretic peptide, NYHA—New York Heart Association, PAD—peripheral artery disease, PCI—percutaneous coronary intervention, RBC—red blood cells, SBP—systolic blood pressure, SGLT2i—sodium-glucose co-transporter 2 inhibitor, SOTA—sotagliflozin, T1DM—type 1 diabetes mellitus, T2DM—type 2 diabetes mellitus, TIA—transient ischemic attack, UACR—urine albumin-to-creatinine ratio, ULN—upper limit of normal, VHD—valvular heart disease, y.o—years old.

## Data Availability

Not applicable.

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
