# Peer review of "SGLT2 Inhibitors in Type 2 Diabetes Mellitus and Heart Failure—A Concise Review"

_jcm, 2022, doi:10.3390/jcm11061470_

Round 1

Reviewer 1 Report

The paper “SGLT2 Inhibitors in Type 2 Diabetes Mellitus and Heart Failure – A Concise Review" by Keller et al. is a good overview on the role of SGLT2 inhibitors in Type 2 Diabetes Mellitus and Heart Failure.

The article is well written. The paper has a good design. The article is logically divided into sections and subsections. The references cited are relevant and adequate. The work has an average degree of novelty and of good interest to the readers.

Comments:

  • Line 43: The tight link between diabetes and heart failure, which should be better underlined in the paper, is represented by the detrimental effect of chronic hyperglycaemia (glucotoxicity) which is mainly mediated by oxidative stress, increased formation of AGEs and enhanced substrate flux through alternative metabolic pathways (DOI: 10.3389/fmed.2021.695792)
  • Only little is presented on the beneficial effect of SGLT2 inhibitors on the endothelial disfunction, which, according to several studies, plays a pivotal role in the pathophysiology of atherogenesis and is usually accompanied by increased oxidative stress and inflammatory responses. (DOI: 10.3390/biomedicines9101356).
  • Line 325-332: In this paragraph the author deals with diabetic ketoacidosis. SGLT2 inhibitors increase plasma ketone levels with an increased risk of diabetic ketoacidosis in type 1 diabetes. However, ketone bodies increased production also represent a good alternative substrate, able to improve the cardiac metabolic efficiency. Some studies on humans and animal models have shown an improvement of the cardiac function and metabolism by beta-hydroxybutyrate (β-OHB), thus inducing reverse ventricular remodelling, with a consequent improvement in the cardiac output and diastolic function (DOI: 3390/ijms22115863).

Author Response

Dear Reviewer,

Thank you for your comments. We find them very valuable.

We cited all the articles you proposed.

Please find the changes regarding your remarks marked in yellow in the attached file.

The text was checked by an English native speaker.

Daria Keller

Reviewer 2 Report

The review is well done and comprhensive. It offers a wide panoramic over the SGLT2i world, focusing on their use both on diabetes mellitus and cardiovascular disease. Some minor revision should be done to improve the manuscript:

1) please define all the abbreviations first time they appear in the text, abstract, tables and figures.

2) some sentences are redundant, please revise the manuscript in order to be more concise. I also suggest to be more concise when discussing trials of SGLT2i application in cardiovascular disease. 

3) An interesting aspect regards the timing of SGLT2i administration across the different heart failure stages (i.e. acute, chronic etc). Please improve the discussion regarding this aspect (see Heart Fail Rev. 2021 Oct 16. doi: 10.1007/s10741-021-10170-1), as well as regarding the protective effects on other organs, which are often involved during the natural history of heart failure (see Int J Mol Sci. 2020 Oct 22;21(21):7833. doi: 10.3390/ijms21217833.). 

4) Fine English editing is required

Author Response

Dear Reviewer,

Thank you for your comments. We find them very valuable.

We cited all the articles you proposed.

Please find the changes regarding your remarks marked in green in the attached file.

The text was checked by an English native speaker. We removed British English, checked the consistency of abbreviations, grammar and flow, as well as removed redundant sentences.

Daria Keller
